# The Effect of Silica Particle Size on the Mechanical Enhancement of Polymer Nanocomposites

**DOI:** 10.3390/nano13061095

**Published:** 2023-03-18

**Authors:** Evagelia Kontou, Angelos Christopoulos, Panagiota Koralli, Dionysios E. Mouzakis

**Affiliations:** 1Mechanics Department, School of Applied Mathematical and Physical Sciences, National Technical University of Athens, Iroon Polytechniou 9, Zografou, 15780 Athens, Greece; 2Institute of Chemical Biology, National Hellenic Research Foundation, 48 Vassileos Constantinou Avenue, 11635 Athens, Greece; 3Hellenic Army Academy, Leoforos Eyelpidon (Varis-Koropiou) Avenue, Vari P.O., 16673 Attica, Greece

**Keywords:** mechanical properties, interfacial properties, polymer–matrix composites, silica

## Abstract

In the present work, SiO_2_micro/nanocomposites based on poly-lactic acid (PLA) and an epoxy resin were prepared and experimentally studied. The silica particles were of varying sizes from the nano to micro scale at the same loading. The mechanical and thermomechanical performance, in terms of dynamic mechanical analysis, of the composites prepared was studied in combination with scanning electron microscopy (SEM). Finite element analysis (FEA) has been performed to analyze the Young’s modulus of the composites. A comparison with the results of a well-known analytical model, taking into account the filler’s size and the presence of interphase, was also performed. The general trend is that the reinforcement is higher for the nanosized particles, but it is important to conduct supplementary studies on the combined effect of the matrix type, the size of the nanoparticles, and the dispersion quality. A significant mechanical enhancement was obtained, particularly in the Resin/based nanocomposites.

## 1. Introduction

It has been proven that impressive mechanical enhancement can be achieved by incorporating inorganic particulate fillers into a polymeric matrix. Potential types of particulate fillers are micro/nano-SiO_2_, glass, Al_2_O_3_, Mg(OH)_2_, and CaCO_3_ particles, carbon nanotubes, and layered silicates [1,2,3,4,5,6,7,8,9,10]. The reinforcing mechanism of polymer particulate composites has been the subject of numerous works in terms of micromechanical models for the evaluation of the elastic constants of the composites with varying volume fraction [10,11,12,13,14]. Simple equations are proposed to analyze the effects of size and density on the number [15], surface area, stiffening efficiency, and specific surface area of nanoparticles in polymer nanocomposites. Moreover, the effect of the nanosize of the nanoparticles, the adhesion between the matrix and nanofiller, and the interphase properties are also examined by introducing a number of equations. Polymer/particulate nanocomposites have been prepared and studied for a variety of properties, focusing mainly on the filler content, while the particle size also varied [16]. Nanocomposites based on polymethyl-methacrylate (PMMA) at a 0.04 volume fraction have also been studied. The particle diameters were 15, 25, 60, 150, and 500 nm. The mechanical properties were studied, and a Young’s modulus increment was detected with decreasing particle size. A systematic study on nanocomposites based on silica nanoparticles [17,18] has revealed that by decreasing the particle size, the composite’s properties are enhanced. Critical analysis of the experimental results and theoretical models of the mechanical properties, such as modulus, strength, and fracture toughness of polymer/particulate micro/nanocomposites, is also available in the literature [19]. Parameters such as filler/matrix adhesion, the filler’s loading and size, modulus, strength, and toughness have been extensively studied. Extended experimental and numerical studies have been performed [20] to define the effect of particle size on the elastic properties (modulus, tensile strength, fracture toughness) of the particulate composites. The size of the nanoparticles varied from macro (0.5 mm) to nano (15 nm) scale. It was discovered that the Young’s modulus of the composite increases as particle size decreases at the nanoscale, and that particle sizes at the microscale have little impact on the composite’s Young’s modulus. It was also discovered that particle size substantially impacts the composite’s tensile strength. Already, at a 1 vol % content, the tensile strength increased with decreasing particle size. The opposite trend could be obtained for 3 vol %.

Various silica grades differing in particle size (micro- versus nanosilica) and surface modification (untreated versus modified surface) have affected interfacial and mechanical properties of compression-molded polypropylene composites, at various filler loadings [21]. The mechanical properties enhancement was extensively studied, with the optimum tensile strength obtained for a silica content range around 2–6 vol %.

In another study [22], silica in the form of raw local natural sand was added to high-density-polyethylene (HDPE) in order to develop a composite material in the form of sheets for possible applications in thin film industries, such as packaging or recycling industries (for example, 3D printing). In general, the majority of the mechanical characterization showed a decrease in their values with the filler addition. However, a notable increase in the toughness and elastic modulus of the composite material was observed with 20 wt% at a 25 μm particle size.

On the basis of an epoxy matrix, the addition of silica nanoparticles (23 nm, 74 nm, and 170 nm) resin has been studied [23]. The effect of silica nanoparticle content and particle size on several mechanical and physical properties was investigated. To the same trend, the microporous poly(L-lactic acid) (PLLA) materials were prepared by uniaxial stretching PLLA/20 wt% silica (SiO2) composites in [24]. The SiO_2_ fillers with a particle size of 15 nm, 200 nm, 500 nm, 1 μm, 2 μm, and 5 μm were selected to explore the effect of the particle size of fillers on the microstructures and physical properties of the microporous PLLA materials.

Based on the number of studies performed on the mechanical enhancement of particulate polymer composites, it can be extracted that the Young’s modulus is instantly improved by the presence of either nano/or micro/scale particles, due to their higher stiffness than that of the polymers [25,26,27]. Hereafter, and regarding the tensile strength, it is strongly dependent on the effective stress transfer between matrix and particles and consequently on the adhesion quality between matrix and filler. It was found that inorganic fillers improved the toughness of thermosetting resins [26]. Summarizing the three main factors responsible for the composite’s mechanical enhancement, namely, filler size, filler loading, and filler/matrix adhesion quality, and on the basis of accessible experimental data and modelling results, it was determined that further experimental research on this topic remains an interesting issue.

The main scope of the present work is the preparation and experimental investigation of polymer composites based on two different polymeric matrices, namely, poly-lactic acid (PLA), a thermoplastic material produced from renewable resources, and an epoxy resin that is a thermosetting one appropriate for industrial applications. Both polymeric materials are reinforced with SiO_2_ particles with varying average diameters from the nano to micro scale at the same weight fraction. The effect of silica size of the nanoparticles on the mechanical enhancement, namely, Young’s modulus, and tensile strength was analyzed, while the effect of the polymeric matrix type utilized on the mechanical properties was also studied. To the authors’ best knowledge, no previous extended research on the polymeric matrices has been done with a large variation of the silica size of the nanoparticles. Therefore, the novelty of the present work lies in both the development of composite materials with lower ecological impact and appropriate for industrial applications as well as in presenting new evidence regarding the competitive mechanisms between particle size effect versus agglomerate formation. The size effect of the particle was counterbalanced by agglomerations, revealing that the micro/nanoparticles’ dispersion formation quality plays a decisive role on the composite’s performance. Consequently, it was shown that the particles’ size effect cannot be studied separately but in combination with other parameters, such as matrix type, adhesion quality, and agglomerate formation.

Scanning Electron Microscopy observations supported the tensile experimental results. The study was further supplemented with finite element analysis regarding the Young’s modulus of the composites and the theoretical modelling of both the nanocomposite’s interphase role and tensile modulus variation.

## 2. Materials and Methods

### 2.1. Materials

Two series of polymer composites were prepared based on two different polymeric matrices, a thermoplastic and a thermosetting one, namely, poly-lactic acid (PLA) and epoxy Resin. To study the effect of filler size on the performance of the composites, silica particles of various sizes in the nanometer and micrometer scale were employed at the same particle loading. The silica powder (SiO_2_) is used as filler in this study in one weight concentration, namely 4 wt%, equivalent to a volume fraction V_f_ = 0.025. In the micro scale, the average diameter of silica particles were chosen to be 1.5, 1.0, and 0.5 μm. All fillers were provided by Alfa Aesar (Kandel, DE). Regarding the nanosized particles, different batches of silica powder with a diameter size ranging between 13–22, 15–35, 18–35, and 55–75 nm, according to manufacturer’s specifications, were used, all provided by Nanografi Nano Technology (Talinn, EST). The nanofillers of diameters of 18–35nm, in particular, were surface treated by a KH550 silane coupling agent.

The PLA used is under the commercial name Ingeo^TM^ Biopolymer 2003D, produced by NatureWorks LLC (Minnetonka, MN, USA), and was kindly supplied by the Greek Company M. Procos S.A. The selected grade 2003D has a density of 1.24 g/cm^3^ and a MFR index equal to 6 g/10 min, measured at 210 °C at a load of 2.16 kg, according to ASTM-D1238-65T. Before use, the material was formed in pellet and dried at 45 °C for a minimum of 2 h in a desiccating dryer. The composites based on the thermoplastic matrix PLA were produced by a melt mixing of the fillers with the PLA matrix material, performed with a Brabender mixer. The temperature was set at 160 °C, and the rotation speed of the screws was 40 rpm. Hereafter, the materials were compression molded at 150 °C, using a thermo-press and a special mold of 1.5 mm thickness.

The epoxy resin used in the present work is the diglycidyl ether of bisphenol A (DGEBA), a liquid epoxy resin commonly used in industrial applications. It is available under the trade name GlassCast 10/50 and is a two-part resin specialized for casting. It is a low-viscosity resin (combined with hardener 500 mPa.s) based on bisphenol A and modified with a reactive diluent. The epoxy/silica composite was produced by uniformly dispersing silica particles in the epoxy resin and then adding a curing agent to the mixture. Before adding the particles, they were heated inside an oven at 60 °C for 30 min in order to evaporate any moisture. First, the silica particles were dispersed using a mechanical mixer (500 rpm for 10 min), followed by ultra-sonication (UP100H, by Hielscher, Teltow, Germany) in high intensity for 20 min. The second step involved adding the curing agent to the mixture at a predetermined portion. The solution was again mixed by ultra-sonication in medium intensity for 10 min. After the dispersion of nano-particles, the solution was placed inside a desiccator under vacuum in order to eliminate any bubbles caused due to the mixing process. At the final step the epoxy/silica solution was poured inside orthogonal moulds and left to cure at ambient temperature. After the curing process and a post-curing at 40 °C, for 4 h, solid plates of epoxy/silica composites were extracted from the moulds and appropriate samples were cut-out using a cutting die attached to a hand press. All types of the specimens manufactured are presented in Table 1.

### 2.2. Tensile Testing

Tensile measurements were carried out with an Instron 1121 type tester (Norwood, MA, USA) at room temperature, according to the ASTM D638. The dumbbell type specimens were of a gauge length of 20 mm, and the applied crosshead was 0.5 mm/min, corresponding to an effective strain rate equal to 4.17 × 10^−4^ s^−1^. A laser extensometer—type cross-scanner by Fiedler Optoelektronik GmbH was used for the deformation measurement. The experimental setup is presented in detail in Ref. [28].

### 2.3. Scanning Electron Microscopy (SEM)

The specimens’ surfaces/morphology were observed in a JEOL 7610F ultra high resolution (JEOL—EUROPE, Nieuw-Vennep, The Netherlands), Schottky Field Emission Scanning Electron Microscope (FE-SEM). The specimens were sputtered with Pd/Au prior to examination using a Quorum SC7620 sputter/coater (Quorum Technologies Ltd., Laughton, East Sussex, UK). Imaging was carried out at an operating range between 1–2 kV.

### 2.4. Dynamic Mechanical Analysis (DMA)

DMA experiments were performed by a Q-800, TA Instruments (New Castle, DE, USA) Dynamic Mechanical Analyzer. The mode of deformation applied was single cantilever, and the mean dimensions of sample plaques were of 12.7 mm in width and 17.5 mm in length with a thickness of 1.5 mm. The temperature range varied from ambient up to 90 °C at a heating rate of 3 °C/min. The temperature-dependent behaviour was studied by monitoring changes in force and phase angle, keeping the amplitude of oscillation constant. Four frequencies 1, 5, 10, and 20 Hz were scanned, and the storage (E’) and loss moduli (E’’) curves versus temperature were recorded.

## 3. Results

### 3.1. SEM Results

SEM images of the fracture surfaces of the composites under investigation are depicted in the following Figures. Ιn Figure 1, the PLA/microcomposites, namely, PLA/0.5, PLA/1.0, and PLA/1.5 are shown in Figure 1a–c, correspondingly.

In Figure 1a,b (PLA/0.5, 1.0), a uniform particle distribution and a small number of particle agglomerates can be observed on a glassy fracture surface. In Figure 1b (PLA/1.0), the fracture surface shows signs of ductility (matrix cavitation around particles), while adhesion appears to be limited as well as the tendency for agglomeration. Matrix crazing as well as cavitation-type yielding is evident. Particle/matrix adhesion appears to be limited. In Figure 1c (PLA/1.5), enhanced matrix ductility, induced by particle cavitational yielding, probably owning to limited particle/size adhesion, can be seen. In addition, some agglomerates are visible. Figure 2 illustrates the SEM images of the PLA/nanocomposites.

In Figure 2a (PLA/13-22), semi-ductile behaviour is observed on the surface of the fractured specimen. Larger (~500nm) and smaller (~100nm) agglomerates are observed. Micro-pores are also evident in the picture. In Figure 2b, (PLA/1535), enhanced matrix yielding is observed at the fracture surface. Large nanoparticles (NPs) agglomerated (∅ > 1 μm) dominate the image, inducing cavitation yielding, whereas free NPs are also observed in the polymer matrix. The mechanism of matrix yielding due to large agglomeration (≅1 μm) is evident. The agglomerates appear to be also ductile. They are composed of enough matrix material enabling them to be partially pulled out of the cavities and elongated, as seen in Figure 2b. Some freely distributed NPs inside the polymer matrix can also be observed. In Figure 2c (PLA/18-35), the fracture surface appears to be brittle. The NPs are mostly found in agglomerates of about 500nm to 1μm. In Figure 2d (PLA/55-75), the fracture surface shows signs of ductility, as it is defined by matrix cavitation around large particle agglomerates. These agglomerated NPs can be observed scattered all around the fracture surface, thus dominating the material fracture response. Nano-crazing crossing areas with non-agglomerated particles can also be observed.

In Figure 3 and Figure 4, the SEM images of the Resin/based composites are illustrated. In Figure 3a (Resin/0.5), multiple crazing runs in the matrix bridging debonded particles and agglomerates. Out of plane shear lips hint at plane strain stress loading seen in the specimen at the micro level. In Figure 3b (Resin/1.0), brittle fracture accompanied by out of plane shear lips is evident again. Agglomerated particles and well-dispersed ones can be observed. In Figure 3c (Resin/1.5), the fracture surface is dominated by exfoliated layers of the polymer matrix. Silica particles appear to be well dispersed and well embedded inside the matrix.

In Figure 4a (Resin/13-22), a very brittle fracture surface can be observed. Some microporous/voids are present on the surface, whereas a large agglomerate of silica NPs can be seen at the center of the picture as well as in the x400 inset view. In Figure 4b (Resin/15-35), large NPs agglomerates (1–3 μm) dominate the brittle fracture surface. These agglomerates are well embedded in the epoxy matrix. In Figure 4c,d (Resin/18-35, Resin/55-75), large NPs agglomerates (1–3 μm) spread over the brittle fracture surface can be observed. These agglomerates are well embedded in the epoxy matrix.

### 3.2. Tensile Results

The tensile stress-strain curves of the PLA/based nano and micro/composites are shown in Figure 5. The Resin/based nano and micro/composites are shown in Figure 6.

All tensile properties and the Young’s modulus enhancement are summarized in Table 2. Following Figure 5 and Figure 6, and with reference to the pure matrices, the initial linear response is followed by yielding, and the subsequent fracture. Resin appears to be more ductile compared with PLA, while all composites exhibit a decreased strain at break due to the presence of rigid particles. All composites appear to have a higher Young’s modulus, with the higher enhancement obtained by the Resin/based nanocomposites. Following the tensile results of Table 2, a quite different mechanical enhancement between PLA and Resin/based nanocomposites or microcomposites is observed. In the PLA/nanocomposites, the highest Young’s modulus increment (27%) is obtained for PLA/18-35, followed by a 20% increment for the PLA/15-35. This result is related to the fact that a special surface treatment has been imposed on the nanoparticle of type 18–35. Hereafter, the reduction of the Young’s modulus in PLA/55-75 can be attributed to both the greater size of the nanoparticles and the formation of agglomerates. No systematic dependence of the Young’s modulus on the nanofiller size is obtained. One reason for this may be the close average diameter size between 13–22, 15–35, and 18–35 nanoparticles. The other reason is the agglomerate formation observed by SEM analysis in all composite materials. Therefore, the expected Young’s modulus increment with decreasing particle size is reversed due to the formation of the agglomerates.

On the other hand, it should be mentioned that the modulus enhancement in the composites of the present work, at the specific filler volume fraction examined, is substantially higher than that obtained in previous works for similar filler loadings [7]. Regarding the microparticles, the higher Young’s modulus is obtained for the smaller microparticle size (PLA/0.5), and, hereafter, the modulus is reduced. As a general trend, given that the agglomeration in composites with nanosilica is observed in a smaller extent than in composites with microfillers (finer dispersion of smaller particles) and the aggregates are smaller in size, the nanocomposites have a higher stiffness than microcomposites in both material series investigated. As also mentioned in [21], higher moduli of the composites with nanosilica than microsilica particles may indicate a pronounced influence of interfacial surface or agglomeration extent.

Regarding yield stress, it is lower for the PLA/nanocomposites when compared with that of the PLA matrix, with the exception of PLA/15-35 material. It appears that the yield stress is almost unaffected by the nanofillers for PLA/13-22 and PLA/15-35, whereas it is reduced for the other particle sizes. The strain at break is always lower than that of the PLA matrix, revealing that the presence of the rigid nanoparticles renders the material more brittle. The tensile strength of all composites under investigation is generally increased compared to that of the polymeric matrix, with the exception of PLA/55-75 and PLA/1.5 specimens. As mentioned in [21], the tensile strength of the composites is differently affected by the incorporation of various fillers into the bulk matrix. It can either (i.) increase or decrease or (ii.) demonstrate no difference.

In the Resin/based micro/nano composites, no systematic dependence of the Young’s modulus on the particle size is observed, either at the nano or in the micro scale. The highest modulus increment is obtained for the Resin/55-75, equal to 60%, and is followed by the Resin/18-35 (34.8%). The enhancement of Resin/13-22 is also comparable to that of Resin/18-35. Referring to the micro composites, the Young’s modulus appears not to be greatly affected by the particles’ average size. The micro-sized/Resin composites exhibit almost the same Young’s modulus value. In addition, the yield stress is always higher than that of the Resin matrix, with the exception of Resin/1.0. Moreover, the strain at the break of all types of composites is much lower than the pure Resin matrix.

It is to be noted that there are contradictory results in the literature as far as the particle size effect on the Young’s modulus is concerned. It has been mentioned in Ref. [7] that the change in the size of micron-scale particles does not have a significant effect on the composite’s modulus [25,29,30,31,32], whereas in [7] the Young’s modulus of the composite further increases slightly with the reduction of particle size from 130 μm to 100 nm. Regarding this trend, it is reported in [20] that the Young’s modulus of a particulate composite is not affected by the size of the particle if it is of micron or larger sizes. When the particle size is in the nanoscale, the composite’s Young’s modulus is enhanced with decreasing particle size. In another work [19], it was found that at low filler volume, fractions up to 0.018 of the particle size actually have no effect on the Young’s modulus. At higher particle contents, the modulus slightly decreases as the particle size increases. When the particle size is about 30 nm, then an obvious effect on the modulus can be observed.

In [23], the addition of silica nanoparticles (23 nm, 74 nm, and 170 nm) to a lightly cross-linked epoxy resin was studied. The Young’s modulus generally increased with the volume fraction of silica nanoparticles. However, no significant effect is observed due to the difference in particle size. The lack of a significant effect on the modulus due to particle size observed in this study is consistent with similar particle-filled epoxy resin studies found in the literature [33]. In our work, the 60% Young’s modulus increment obtained in Resin/55-75 was higher than the one obtained for the epoxy resin in ref. [24] at a similar nanofiller volume fraction.

Within the same context, in [34,35], the nanoscale effect on the effective properties of materials reinforced with spherical particles has been examined. The particle size effect on the tensile modulus of the nanocomposite has been investigated. Particle diameters of about 15, 35, and 60 nm were studied. A 9% modulus increment was obtained when the particle diameter reduced from 35 to 15 nm. This trend was not confirmed for larger particles due to the agglomerates formation.

Regarding the yield stress, as mentioned in [23], the yield stress values for a filled epoxy remain constant in silica nanoparticle content as well as particle size. Contradictory results about the yield stress increment or the lowering in particulate composites were reported. In order to explain these observations, the interfacial strength or adhesion between the particles and the epoxy has to be considered, particularly for nanosized particles due to their high surface area of an interface. The poor filler-matrix adhesion leads to a composite unable to carry any part of the external load, resulting, in this way, in a lower tensile strength. Following Table 1, the yield stress of the Resin/based composites is higher than that of the pristine Resin, whereas the failure stress was found to be higher or lower from the tensile strength of the Resin matrix.

On the basis of previous studies, for instance [22], a tensile strength lowering was noticed for the composite materials developed from both 25 μm and 5 μm size filler particles. Such a decreasing trend can be explained by the particles randomly arranging themselves in a way that limits the stress transfer [36].

Moreover, the decreased tensile strength also causes increased brittleness, which can be attributed to the formation of agglomerates. These agglomerates form initiation spots of stress concentrations that lead to failure. Moreover, the formation of voids in matrices is also reported to contribute towards decreased strength values [37,38].

This is in accordance with our results for the Res/55-75, which exhibits the higher tensile strength and the higher failure strain.

The tensile strength increment is associated with the satisfactory adhesion of nanofillers to the matrix, resulting in sufficient stress transfer across the interphase [39]. As it was found in [20], the tensile strength of particulate composites can be increased with decreasing particle size. However, due to the poor dispersion of the particles, which is more severe at higher filler loadings, a threshold of 0.03 particle volume fraction was found, above which a lower tensile strength is obtained. The ultimate strength strongly depends on the stress transfer [7] between the particles and the matrix, when good adhesion is established between them. The imposed stress is then effectively transferred from the matrix to the inclusions [31]. The ultimate strength was not improved for particles larger than 1 μm. Generally, tensile strength increases with the increasing surface area of the filler particles through a more efficient stress transfer mechanism. For this reason, the higher contact surface that is developed in small particles results in tensile strength augmentation. This effect seems to be valid in both PLA and Resin/nanocomposites, with the exception of PLA/55-75 and Res/15-35.

It is well known that the obtained augmentation of the elastic properties is strongly related to several parameters, such as good quality dispersion of nanofillers and proper adhesion between matrix and nanofillers, which are also associated with the possible surface treatment of the nanofillers [40]. It can therefore be assumed that an improved interfacial interaction between particles and matrix accelerates the stress transfer from matrix to nanofillers, resulting in enhanced strength and modulus by dissipating more pull-out energy. In the work by Blivi et al. [16], the effect of silica particle size was experimentally investigated in PMMA reinforced with silica nanoparticles at a 0.04 volume fraction. The Young’s modulus increased with decreasing particle size from 500 to 25 nm. It should be mentioned, however, that a slight modulus increment is obtained for particle sizes varying from 500 to 150 nm. This increment is more obvious for particle sizes ranging from 60 to 15 nm. On the other hand, the highest modulus increment of PMMA was recorded to be of the order of 60% at a volume fraction almost two times higher than that of the present work. In our study, the Resin/nanocomposites modulus increment between 35% and 60% was recorded at a volume fraction equal to 0.025.

Therefore, in spite of the formation of agglomerates, a satisfactory mechanical enhancement was achieved in the composites under investigation, with the particle size assuming a decisive role.

The yield stress may provide additional information on filler–polymer matrix interactions at the stage before plastic deformation is generalized into the bulk material. On the basis of the empirical equation by Turcsányi et al. [41], the extent of interfacial interactions between polymeric matrix and fillers can be estimated by the interaction parameter B, included in the following equation:(1)$$\sigma_{y_{c}}=\sigma_{y_{m}}\frac{{1-V}_{f}}{1+2{.5V}_{f}}{\mathrm{Exp}[\mathrm{BV}}_{f}]$$
where $\sigma_{y_{c}}$ and $\sigma_{y_{m}}$ are the yield stress of the composite and the matrix, correspondingly, and $V_{f}$ is the filler volume fraction. The results of interaction parameter B values, which are calculated on the basis of the yield stress σ_y_ values after Equation (1), are presented in Table 3. Higher B values are exhibited by PLA/13-22 and PLA/15-35 from the PLA series. On the other hand, the epoxy Resin-based composites demonstrate high B values, particularly the Res/15-35 and Res/18-35, as well as the Res/0.5 composite. Parameter B seems to be affected by a number of factors, such as the filler’s surface modification, matrix type, particle size, and dispersion quality, but not in a regular way.

The high parameter B values appear to be in accordance with the corresponding Young’s modulus increment in the Resin/based composites.

The abovementioned results consist of an additional indication that the particle size effect cannot be studied separately but only in combination with other parameters, such as matrix type, adhesion quality, and agglomerate formation.

### 3.3. Dynamic Mechanical Analysis (DMA)

The master curves of the storage modulus E′ of all composites examined were constructed by applying the time-temperature superposition principle and are shown in Figure 7 for all of the composites under investigation. The reference temperature of PLA/based composites is 60 °C, and for the Resin/based ones is 47 °C, both close to the Tg of the polymers.

In all material types, the reinforcing effect of the silica particles is revealed. In Figure 7a and in the high-frequency range (glassy state), it can be observed for the micro/composites that the PLA/1.0 composite attains the highest storage modulus followed by the PLA matrix and the PLA/0.5 and PLA/1.0 microcomposites. In the low-frequency range (rubbery state), the storage modulus differences are more prominent, while the trend is not similar to the one in the tensile results. All PLA microcomposites exhibit the so-called softening dispersion—that is, the height difference between the glassy and rubbery state, which constitutes an indication of a more elastic behaviour, due to the presence of rigid silica particles. In the low-frequency range, the PLA/1.5 and PLA/0.5 demonstrate a plateau, which can be associated with the formation of a network between matrix and silica particles. On the other hand, a rubbery plateau is obtained for all PLA/nanocomposites, revealing that the nanoparticles facilitate the formation of a network. The highest reinforcing effect and the lowest height difference are exhibited by the PLA/55-75. The non-monotonic dependence on the silica size of the nanoparticles in the low-frequency range can be associated with the general complex morphology and the existence of regions with different matrix/particles connection [27]. Therefore, localized regions may have the same stress or the same strain upon an external stress field. At the low frequencies, with the response registered over a long period, the polymeric structure has the stronger contribution, revealing a decreased storage modulus [27]. In Figure 7c,d, the Resin/micro and nano-composites have almost the same storage modulus in the glassy state, whereas all Resin composites appear to have the rubbery plateau due to both the chemical crosslinks and the network developed between the matrix and silica particles, indicating the structure of the system.

### 3.4. Modeling of the Effective Modulus Tensor

#### 3.4.1. Finite Element Analysis (FEA)

In this section, the effective elastic modulus of the PLA and Resin/based nanocomposites/microcomposites has been analyzed by a 3D finite element method analysis (FEA).

Different 3D representative volume elements (RVEs) of PLA/SiO_2_ and Resin/SiO_2_ composites at the various particle sizes under investigation have been generated using Digimat-FE. The RVE generation included incorporated spherical with agglomeration and also spherical with agglomeration and interphase SiO_2_ particles, and their elastic modulus was evaluated using the Digimat-FE solver. After generating the RVE, it was exported to a Digimat FE solver by defining material properties, definition of boundary conditions, etc. to evaluate the resulting RVE elastic modulus using the finite element method. The matrix was treated as an isotropic material with a Young’s modulus equal to 3000 MPa for PLA and 2150 MPa for the Epoxy Resin. The Poisson’s ratio was equal to 0.28 for both matrices. The SiO_2_ particles were also treated as isotropic materials with a Young’s modulus equal to 70,000 MPa and a Poisson’s ratio equal to 0.26. When the existence of an interphase between matrix and silica particles is assumed, the interphase’s modulus was taken equal to 20,000 MPa. In conventional composites, the RVE typically consists of a low number of reinforcing particles enclosed by matrix, and by applying appropriate boundary conditions to it, the effect of adjacent materials is covered. However, in the nanocomposites, due to the difference in the dimensions of the silica nanoparticles and matrix, the number of nanoparticles in the RVE is much more than one reinforcing part [42]. In our study, when the SiO_2_ dimensions are in the nanoscale, a large number of nanoparticles should be included within the matrix for a correct RVE generation. The representative element is formed by entering the input parameters, including the geometric characteristics of the silica particles, the weight fraction of the nanotube, the elastic modulus of each phase, etc., in the Digimat-FE software. Representative volume elements (RVEs) of both procedures, without and with interphase, are illustrated in Figure 8, representatively, for 15–35 nm and 1.0 μm.

The RVEs elastic moduli were obtained and presented in Table 4.

When the silica particles and their agglomeration are considered, the agreement with the experimental results of the Young’s modulus is not very satisfactory. The FEA results, with the assumption of an interphase between silica particles and the matrix, exhibit a very satisfactory agreement between FEA results and the experiment. This result is strongly related to the adhesion quality effect, which has been proven to be decisive in mechanical enhancement. In Figure 9, the variation of the effective modulus with varying the relative interphase volume fraction is depicted for all types of composites examined. The relative interphase volume fraction appropriate to model the experimental data is shown in Figure 9.

From the FEA results of Figure 9, it can be concluded that the Young’s modulus increases with decreasing particle diameter as well as with increasing interphase volume fraction. The points in Figure 9 are the FEA results obtained at the specific interphase volume fractions and are very close to the experimental data. The deviation between experiments and FEA/interphase results is relatively low, as shown in Table 4. No specific trend of the interphase volume fraction is observed for the various nano or micro/composites. It can be noticed, however, that for both PLA and Resin/based nanocomposites (with the exception of PLA/55-75), the highest interphase volume fraction was estimated for the 18-35 silica nanoparticle. This is in accordance with the specific coating imposed on these nanoparticles.

#### 3.4.2. Analytical Model

It was considered to be straightforward that the Young’s modulus of the composites under investigation would also be evaluated on the basis of a micromechanics-based model for the elastic stiffness tensor—presented in [43] and previously introduced in [44]. In [43], the nanocomposite is considered to contain randomly located spherical nanoparticles, while a spherical nanoparticle is coated by a graded interphase layer, consisting of the so-called effective particle. The elastic stiffness tensor of the nanocomposite is given by:(2)$$\bar{C}=C^{0}•\left[I-\Phi^{\Sigma }T^{\Sigma }•{\left(\Phi^{\Sigma }S•T^{\Sigma }+I\right)}^{-1}\right]$$
where **C^0^** is the matrix stiffness tensor, **I** the identity tensor, **Φ^Σ^** the effective particle volume fraction, **S** is the Eshelby tensor, and **T^Σ^** is a tensor given by:(3)$$T^{\Sigma }=\Phi^{Ρ/\Sigma }T^{P}+\Phi^{I/\Sigma }T^{I}$$
where **Φ^Ρ/Σ^** and **Φ^Ι/Σ^** are the volume fraction of nanoparticle and interphase inside the effective inclusion. Tensor **T^I^**, which, according to the double-inclusion method of Hori and Nemat-Nasser [45], is expressed as:(4)$$T^{I}=-{\left(S+A^{I}\right)}^{-1}$$
with tensor **A^I^** defined as:(5)$$A^{I}=\left(C^{I}-C^{0}\right)•C^{0}$$
**C^I^** is the elastic stiffness tensors of the interphase.

The components of the Eshelby tensor **S** for an isotropic spherical particle in an isotropic matrix are given by:(6)$$\begin{array}{l} S_{11}=S_{22}=S_{33}=\frac{7-5\nu_{0}}{15\;(1-\nu_{0})}\\ S_{12}=S_{23}=S_{31}=-\frac{1-5\;\nu_{0}}{15\;(1-\nu_{0})}\\ S_{44}=S_{55}=S_{66}=\frac{4-5\;\nu_{0}}{15\;(1-\nu_{0})} \end{array}$$
where ***ν*_0_** is the Poisson’s ratio of the matrix.

According to the geometry of the particle surrounded by the interphase (Figure 1a), the effect of nanoparticle size can be explicitly introduced on the elastic stiffness tensor, by expressing the quantities **Φ^Σ^**, **Φ^Ρ/Σ^**, and **Φ^Ι/Σ^** as follows:(7)$$\begin{array}{l} \Phi^{Ρ/\Sigma }=\frac{r_{p}^{3}}{{{(r}_{p}+e)}^{3}}\\ \Phi^{I/\Sigma }=1-\frac{r_{p}^{3}}{{{(r}_{p}+e)}^{3}}\\ \Phi^{\Sigma }=\Phi^{P}{\left(1+\frac{e}{r_{p}}\right)}^{3} \end{array}$$
where **Φ^Ρ^** is the particle volume fraction, **e** is the interphase thickness, and **r_p_** is the nanoparticle radius.

The Young’s modulus experimental results of the Resin/based silica composites were employed to implement the analytical model. Following the above equations, the elastic stiffness tensor $\bar{C}$ can be evaluated with the required parameters of the elastic properties of the interphase, and, consequently, the tensile modulus of the composite can be calculated. Poisson’s ratio of the Resin matrix was taken equal to *v_0_ =* 0.23, whereas the particles’ Poisson’s ratio was taken equal to 0.23, while the modulus of silica nanoparticles was taken equal to 70 GPa. The interphase modulus **E_i_** was varied between the matrix and silica nanoparticles’ moduli and was taken to be equal to 20GPa, similar to that utilized in the FE analysis. The interphase thickness is an additional parameter in the analytical model. In this analysis, the experimental results of the Resin/based composites were utilized. Calculations at various values of interphase thickness were performed to approximate the experimental values of Young’s modulus. In Figure 10, the Young’s modulus model calculated values with varying silica particles’ size and varying interphase thicknesses, which are depicted together with the experimental data of the Resin/based composites. More specifically, model calculations were made for three different interphase thickness e values, namely 5, 10, and 20 nm. It is shown that in the micrometer scale, the Young’s modulus is not significantly affected, exhibiting the same value at the three different SiO_2_diameters. Therefore, and for the reason of clarity, only the results for a diameter of 500nm are presented. Within the frame of the analytical mode, when selecting the interphase thickness, the interphase volume fraction $\Phi^{I/\Sigma }$ could be hereafter evaluated on account of Equation (7) for the best approximation with the experimental Young’s modulus data. In Table 5, the calculated values of the interphase volume fraction, estimated by the analytical model, are presented comparatively with the ones obtained by the FE analysis.

It has to be noted that high interphase volume fractions were calculated by the analytical model as far as the Resin/nanocomposites is concerned. High interphase volume fractions have been found in previous works, where it was assumed that the interphase volume fraction can be equal to half of the matrix volume fraction [10]. This is an assumption close to the results of molecular modeling performed in the work by Odegard et al. [46]. The presence of many polymer chains at the interphase means that much of the polymer is ‘interphase-like’ [46,47]. This is not the case for micron-sized silica particles, where the interphase volume fraction is very low. From Table 5 and comparing the interphase volume fraction results, it can be extracted that a lower deviation between FEA and the analytical model is obtained for the micron-sized SiO_2_ particles. This result can be attributed to the fact that the analytical model works better in the case of larger particles.

## 4. Conclusions

In the present work, SiO_2_ composites based on PLA and epoxy Resin were prepared and experimentally studied. The silica particles were of varying size—from the nano to μm scale at the same weight fraction. Regarding PLA/composites, the highest mechanical improvement was exhibited by the PLA reinforced with silica nanoparticles of an average diameter equal to 0.025 (PLA/15-35) and 0.0265 nm (PLA/18-35). A Young’s modulus improvement was also obtained for PLA reinforced with silica particles with an average diameter of 0.5 μm. This is in accordance with SEM images, where uniform agglomerates of a moderate size of the order of 500 nm have been noticed.

An impressive mechanical enhancement has been obtained in the Resin/silica nanocomposites, substantially higher than the one in the PLA/nanocomposites at the same SiO_2_ weight fraction. This mechanical improvement is also superior to that achieved in previous works. The specific epoxy resin appears to have better adhesion with the dispersed silica nanoparticles, whereas the mechanical improvement of the Resin/micro composites is very low at the same silica loading. In addition, an improvement of the yield stress was also obtained. These results are compatible with FE analysis, which premise an increased relative interphase volume fraction compared to that in the PLA/composites. In addition, the calculated interphase volume fraction for the microcomposites was quite low, which is in accordance with SEM observations, considering both the particles’ size and the interphase. In addition, a widely known analytical model was implemented, considering both the particles’ size and the interphase. Comparing the interphase volume fraction estimated by the two models, a lower deviation was obtained for the composites reinforced with silica particles at the micron scale.

The enhancement findings of the present research reveal the combined effect of the matrix type and the size of the particles on the composite’s overall properties. Additionally, it has been postulated that the effect of particles’ size on the mechanical and thermomechanical properties cannot be separately evaluated, since it is usually counterbalanced by the agglomeration’s formation, revealing that the micro/nano-particles’ dispersion, in combination with the matrix type and filler adhesion quality, plays a decisive role on the performance of the composites.

## Figures and Tables

**Figure 1 nanomaterials-13-01095-f001:**
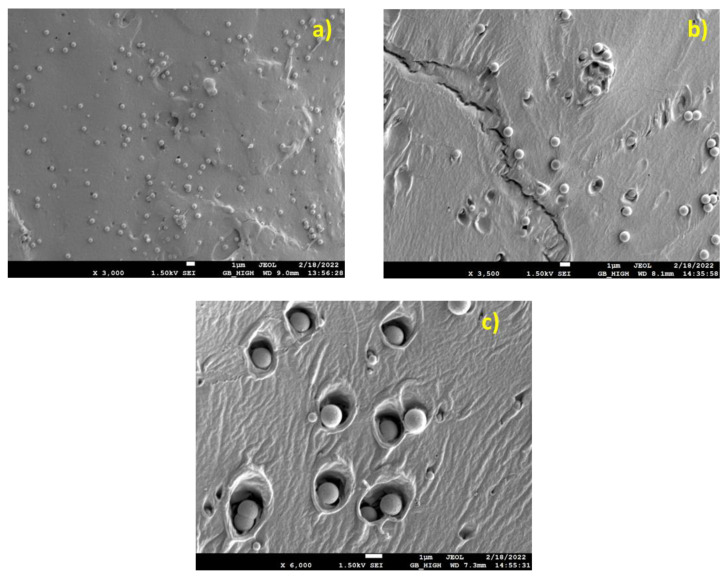
SEM micrographs of PLA/SiO_2_ micro-composites, (**a**) PLA/0.5, (**b**) PLA/1.0, (**c**) PLA/1.5.

**Figure 2 nanomaterials-13-01095-f002:**
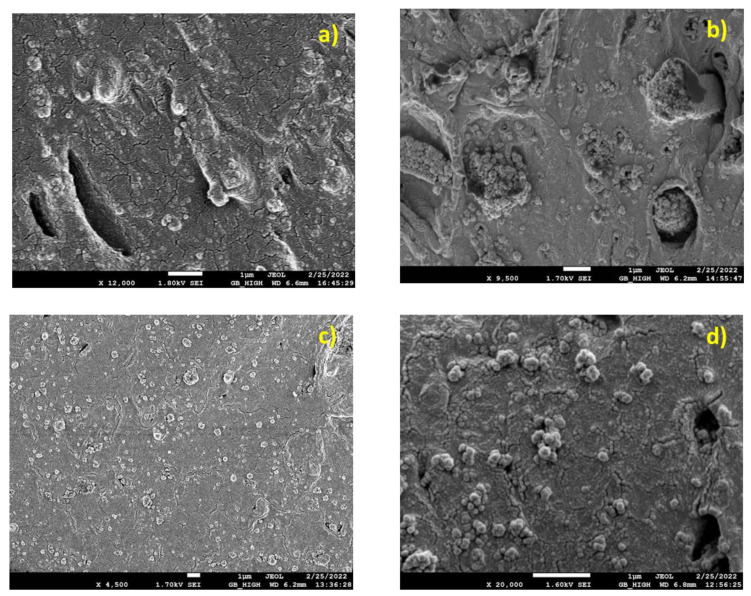
SEM micrographs of PLA/SiO_2_ nano-composites, (**a**) PLA/13-22, (**b**) PLA/15-35, (**c**) PLA/18-35, (**d**) PLA 55-75.

**Figure 3 nanomaterials-13-01095-f003:**
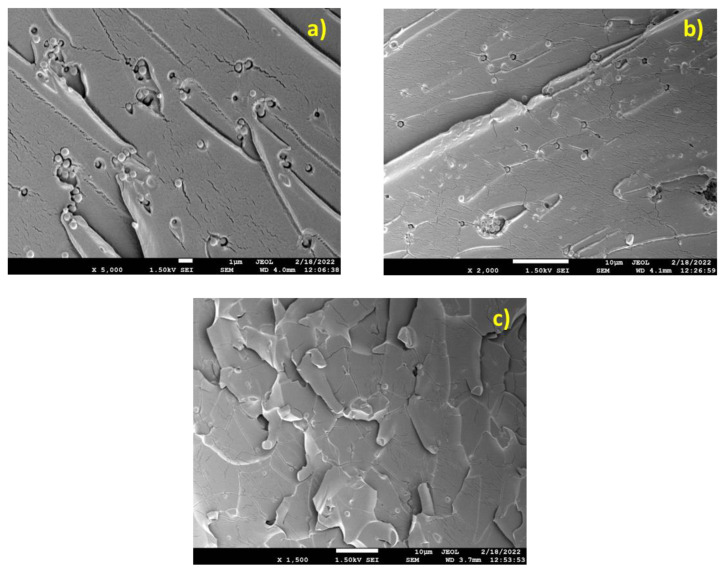
SEM micrographs of Resin/SiO_2_ micro-composites, (**a**) Resin/0.5, (**b**) Resin/1.0, (**c**) Resin/1.5.

**Figure 4 nanomaterials-13-01095-f004:**
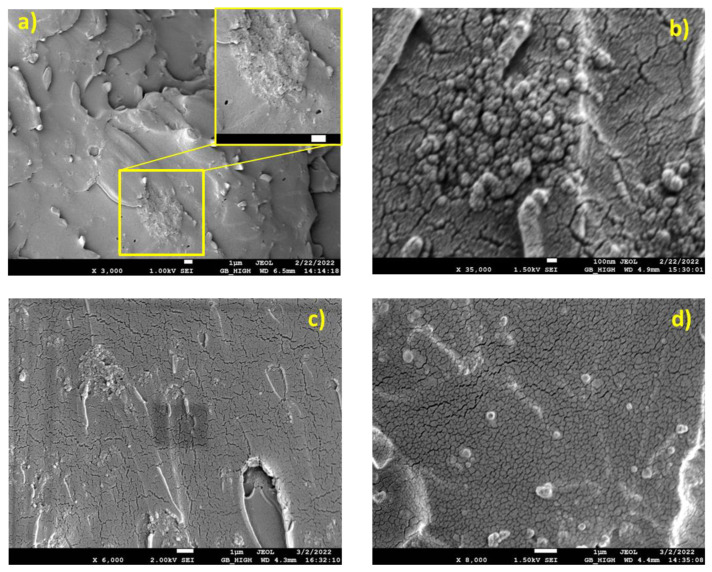
SEM micrographs of Resin/SiO_2_ nano-composites, (**a**) Resin/13-22, (**b**) Resin/15-35, (**c**) Resin/18-35, (**d**) Resin 55-75.

**Figure 5 nanomaterials-13-01095-f005:**
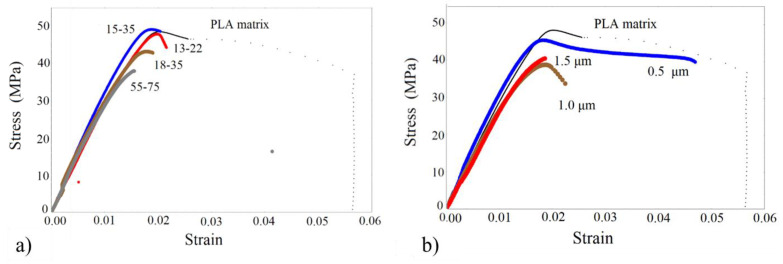
Tensile stress-strain curves of (**a**) PLA/silica nanocomposites, (**b**) PLA/silica micro-composites.

**Figure 6 nanomaterials-13-01095-f006:**
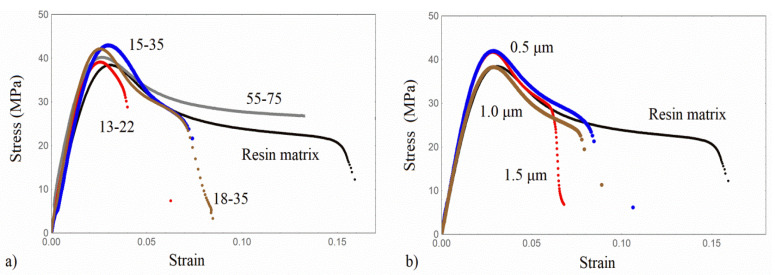
Tensile stress-strain curves of (**a**) Resin/silica nanocomposites, (**b**) Resin/silica micro-composite.

**Figure 7 nanomaterials-13-01095-f007:**
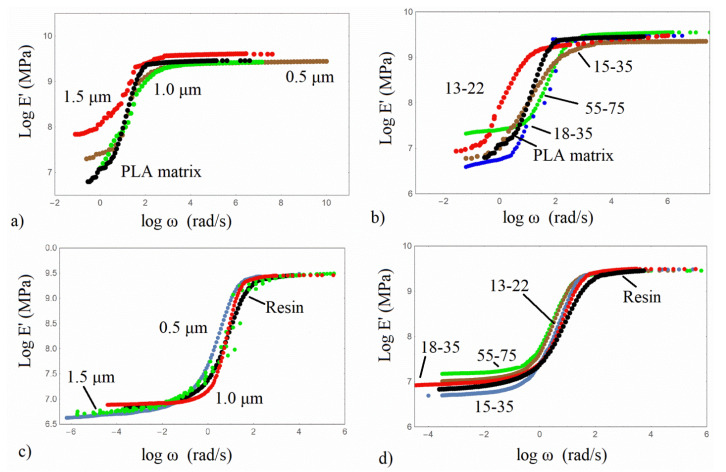
Master curves of storage modulus of the composites examined. (**a**,**b**) PLA/based silica composites, (**c**,**d**) Resin/based silica composites.

**Figure 8 nanomaterials-13-01095-f008:**
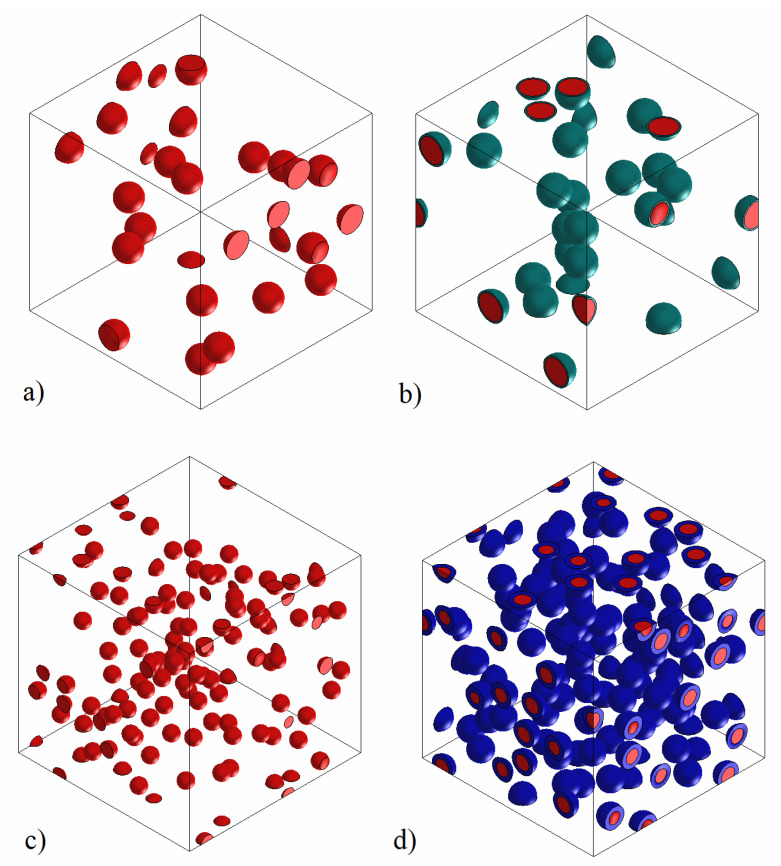
Representative RVEs of the composites examined: (**a**) PLA/1.0 (**b**) PLA/1.0/interphase (**c**) PLA/15-35 (**d**) PLA/15-35/interphase.

**Figure 9 nanomaterials-13-01095-f009:**
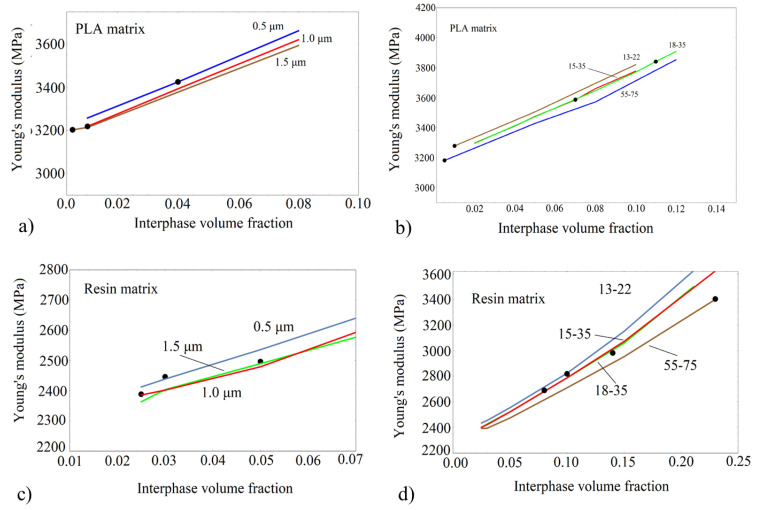
Young’s modulus by FE analysis with varying relative interphase volume fraction for all SiO_2_ composites examined. Lines: Finite Element simulation; Points: Experimental data (**a**) PLA/SiO_2_ with particle’s diameter at the micro-scale, (**b**) PLA/SiO_2_ with particle’s diameter at the nano-scale, (**c**) Resin/SiO_2_ with particle’s diameter at the micro-scale, (**d**) Resin/SiO_2_ with particle’s diameter at the nano-scale.

**Figure 10 nanomaterials-13-01095-f010:**
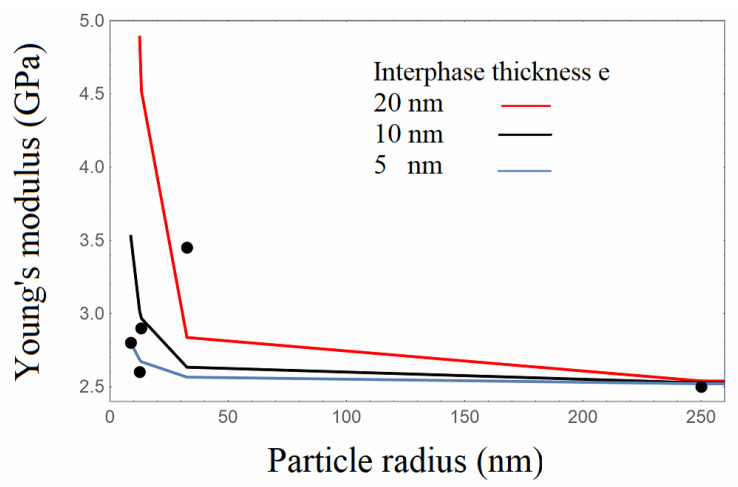
Young’s modulus with varying particle radius, at various values of the interphase thickness. Points: Experimental results, Lines: Analytical model.

**Table 1 nanomaterials-13-01095-t001:** Micro- and Nano-composites Manufactured and Related Specimen’s Designations.

Specimen’s Designations	Matrix Type	SiO_2_Filler wt%/Vf	SiO_2_Filler Diameter∅ [nm]
PLA	poly-lactic acid (PLA)	0/0	-
PLA/13-22	-//-	4/0.025	13–22
PLA/15-35	-//-	-//-	15–35
PLA/18-35 *	-//-	-//-	18–35
PLA/55-75	-//-	-//-	55–75
PLA/0.5	-//-	-//-	500
PLA/1.0	-//-	-//-	1000
PLA/1.5	-//-	-//-	1500
Resin ES-35	bisphenol A—Epoxy- DGEBA	0/0	-
Res/13-22	-//-	4/0.025	13–22
Res/15-35	-//-	-//-	15–35
Res/18-35 *	-//-	-//-	18–35
Res/55-75	-//-	-//-	55–75
Res/0.5	-//-	-//-	500
Res/1.0	-//-	-//-	1000
Res/1.5	-//-	-//-	1500

* silanized particles.

**Table 2 nanomaterials-13-01095-t002:** Tensile properties of the composites examined.

Material/ Specimens	Young’s Modulus (MPa)	Modulus Increment%	Yield Stress (MPa)	Yield Strain	Tensile Strength (MPa)	Failure Strain
PLA	3000 ± 120	-	48.6 ± 3.5	0.02	36.9	0.056
PLA/13-22	3200 ± 140	7	47.8 ± 3.1	0.019	43.3	0.02
PLA/15-35	3600 ± 144	20	48.9 ± 3.0	0.018	48.6	0.02
PLA/18-35	3800 ± 185	27	43.0 ± 2.8	0.016	43.0	0.016
PLA/55-75	3202 ± 135	-	37.8 ± 2.8	0.015	31.0	0.04
PLA/0.5	3455 ± 131	15	45.8 ± 2.8	0.017	41.3	0.047
PLA/1.0	3250 ± 130	8	39.2 ± 2.4	0.018	32.0	0.025
PLA/1.5	3120 ± 123	4	41.0 ± 2.3	0.019	-	0.018
Resin ES-35	2150 ± 107	-	38.5 ± 3.2	0.03	20.0	0.17
Res/13-22	2800 ± 133	30	40 ± 3.3	0.024	22.5	0.07
Res/15-35	2600 ± 117	21	43 ± 2.1	0.028	20.0	0.09
Res/18-35	2900 ± 122	34.8	43 ± 3.0	0.028	25.0	0.08
Res/55-75	3450 ± 149	60	40 ± 2.1	0.027	26.0	0.13
Res/0.5	2470 ± 104	15	42.2 ± 2.8	0.028	20.0	0.085
Res/1.0	2380 ± 100	11	38 ± 4.0	0.028	23.6	0.08
Res/1.5	2500 ± 108	16	41.5 ± 2.5	0.026	25.0	0.07

**Table 3 nanomaterials-13-01095-t003:** Interaction parameter B values.

Material	Interaction Parameter
PLA	-
PLA/13-22	2.77
PLA/15-35	3.68
PLA/18-35	-
PLA/55-75	-
PLA/0.5	1.06
PLA/1.0	-
PLA/1.5	-
Resin	-
Res/13-22	4.96
Res/15-35	7.86
Res/18-35	7.86
Res/55-75	4.96
Res/0.5	7.10
Res/1.0	2.91
Res/1.5	6.44

**Table 4 nanomaterials-13-01095-t004:** Finite element analysis results.

Material	Average Particle Diameter	Young’s Modulus Experim.	Young’s Modulus FEA	FEA Results Deviation from Experim.	Young’s Modulus FEA/Interphase	FEA/Interphase Results Deviation from Experim.
	(nm)	(MPa)	(MPa)	(%)	(MPa)	(%)
PLA	-	3000	-	-	-	-
PLA/13-22	17.5	3200	3128	2.25	3284	2.62
PLA/15-35	25.0	3600	3131	13.0	3585	0.41
PLA/18-35	26.5	3800	3197	15.8	3840	1.0
PLA/55-75	65.0	3202	3162	1.25	3226	0.75
PLA/0.5	500	3455	3205	7.23	3425	0.10
PLA/1.0	1000	3250	3177	2.25	3219	0.95
PLA/1.5	1500	3120	3161	1.31	3180	1.90
Resin ES-35	-	2150	-	-	-	-
Res/13-22	17.5	2800	2321	17.1	2819	0.67
Res/15-35	25.0	2600	2300	11.5	2690	3.46
Res/18-35	26.5	2900	2297	20.8	2982	2.82
Res/55-75	65.0	3450	2278	33.9	3400	1.45
Res/0.5	500	2470	2273	7.97	2446	0.97
Res/1.0	1000	2380	2280	4.2	2388	0.33
Res/1.5	1500	2500	2275	9.0	2496	0.16

**Table 5 nanomaterials-13-01095-t005:** Interphase volume fraction calculated by FEA and the analytical model.

Material	Interphase Volume Fraction (FEA)	Interphase Volume Fraction (analytical Model)
Resin	-	-
Res/13-22	0.10	0.74
Res/15-35	0.08	0.63
Res/18-35	0.14	0.81
Res/55-75	0.23	0.88
Res/0.5	0.03	0.018
Res/1.0	0.025	0.017
Res/1.5	0.05	0.01

## Data Availability

Not applicable.

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
