# Peer review of "The Effect of Silica Particle Size on the Mechanical Enhancement of Polymer Nanocomposites"

_nanomaterials, 2023, doi:10.3390/nano13061095_

Round 1

Reviewer 1 Report

In this paper, the Authors study the influence of silica particle’s size on the mechanical properties of polymer nanocomposites. Two host polymers are studied: poly-lactic acid (PLA) and epoxy resin (diglycidyl ether of bisphenol A, DGEBA). The distribution of particles is studied using the scanning electron microscopy (SEM). The main mechanical properties of the nanocomposites are studied: Young’s modulus, yield stress, yield strain, tensile strength, and failure strain. The dynamic mechanical analysis (DMA) was also preformed. The general trend is that the reinforcement is higher for the nanosized particles than that of micrometer-sized particles. The experimental results are compared with the finite element model and the analytical model taking into account the presence of the interphase.

The experimental results are of interest in the field of nanocomposite research and the study of reinforcement by nanoparticles.

The experimental results are consistent with the following generally accepted view. The presence of stiff nanoparticles in the host material leads to the increase of the overall stiffness of the nanocomposite. Due to the surface interaction between the host material of the nanoparticle there is the interphase of some thickness around nanoparticles, which further increase the overall stiffness of the nanocomposite. The thickness of the interphase are determined by some microscopical properties, like the thickness of the adhesion layer. Therefore, the effect of the presence of nanosized particles is higher than that of micrometer-sized particles for the same volume concentration of particles.

However, I cannot agree with the following point of view: in the manuscript, the difference in the stiffness for composites with different particle size is attributed only to the presence of the interfase. The main problem is the precision of the obtained stiffness (see Table 2). For example, the difference, between 3000±120 and 3200±140 is close to the precision. Therefore, there are uncontrollable fluctuations, which could not be attributed to the interfase. Also, the mechanical properties of polymers (and polymer composites) may depend on many details of the manufacturing process. As a result, there are a lot of factors except the interfase, which also may affect the stiffness. Therefore, it is not correct to find the interfase properties, which could compensate all these fluctuations.

It is especially important the micrometer-size particle. The proposed analysis leads to the presence of the interfase of the micrometer thickness, which is an extremely thick interfase. So I propose to reconsider the analysis. At the same time, I note that this is not a drawback of the experiment, since further improvement of accuracy is a very laborious task. Probably, one can find the constant interface thickness across all samples, which gives the best fit for all obtained moduli (maybe except the silanized particles).

In addition to the major comment above, there are the following comments:

1) The density of PLA and DGEBA is a bit different. However, for both of them there is 4%wt and 0.025 volume fraction of nanoparticles, which is possible for the same density only. It would be better to take into account the actual polymer density to increase the precision.

2) Table 2: column “Modulus Increment” is shifted by half a row.

3) Table 3: The interaction parameter is missed for some composites.

4) Table 4. Column 4 shows the Young’s modulus obtained by the finite element analysis without the interfase. The presented values fluctuate between 3128 MPa to 3205 MPa. However, without the interfase, it is the scale-free model so all the values should be the same. Probably, such fluctuations indicate the precision of the finite element analysis caused by an insufficient number of finite elements.

5) Table 4. Last column shows the deviation from experiment in the model with the interfase. However, it is always possible to find the interfase thickness to obtain zero error (except the case of PLA/1.5 whose Young’s modulus is smaller than the prediction without the interfase).

6) Figure 10: It is not entirely clear what conclusion can be drawn from this figure.

7) Table 5: The presented interphase volume fractions have enormous values (up to 0.5).

To conclude, a major revision of the data analysis is necessary to publish the manuscript in Nanomaterials.

Author Response

Dear Editor,

Thank you for sending us the reviewer’s comments. We appreciate their time and effort based on which we have improved our work.

I am sending you the revised manuscript with the proper corrections and explanations. Please observe the highlighted text.

As follows, there is a point to point response to the reviewer’s suggestions.

We hope that both reviewer’s comments are addressed and the revised manuscript meets the standards of the Journal Nanomaterials.

Thank you for your efforts and help

Best regards

Dionysios Mouzakis

Reviewer 2 Report

This paper studies the tensile properties of SiO2 composites based on poly-lactic acid (PLA) and epoxy resin. It is positive that the authors adopted the experiment, Finite element analysis, and analytical model to carry out. Some issues that need to be settled are listed as follows before acceptence:

1.      In the abstract, “The thermomechanical performance of the composites prepared was studied in combination with scanning electron microscopy (SEM).”, but I didn’t see the relevant explanation in the article. So, I want to know which thermodynamic properties have been studied. In addition, in conclusion, “it is difficult to evaluate the effect of particle’s size alone on the thermomechanical properties”, this seems to be contradictory to the abstract.

2.      Finite element analysis (FEA) has been performed to analyze Young’s modulus of the composites, why were yield stress, tensile strength in Table 2 didn’t analyzed?

3.      In Page 137, “Before adding the particles, both Resin and curing agent, were heated inside an oven at 60 ℃ degrees for 30 min, in order to evaporate any moisture.”, is this sure to work as an evaporate? And the symbol “℃” is wrong.

4.      In Page 227, how a very brittle fracture surface is observed and what is the reason for determining?

5.      In Page 286-288, the authors pointed out that the presence of the rigid nanoparticles render the material more brittle, while the SiO2 and Resin are not in close contact, there are gaps, cracks, and holes in Figure 3. So, I have some skepticism of this conclusion.

6.      In Finite Element Analysis, the interphase’s modulus was taken equal to 20000 MPa, it is suggested that the authors explain the basis for determining the interfacial’s modulus and the interface properties.

7.      In Conclusions, “the calculated interphase volume fraction for the microcomposites was quite lower, which is in accordance with SEM observations.”, “considering both the particle’s size and the interphase”. It is suggested that the author add the appropriate explanation to the text.

Author Response

(The authors gave the same response as above.)

Round 2

Reviewer 1 Report

Please find comments in the attached file

Author Response

 Dear Editor,

Thank you again for sending us the reviewer’s comments. We are thankful to him too for his accurate observations.

Following these comments, we were able to perform the proper corrections in our work. I am sending you the revised manuscript.

In the following text, there is a point to point, response to the reviewer’s suggestions.

We believe that all reviewer’s comments have been properly addressed. We are looking forward to your decision on our work.

FIRST Query

I would emphasize, that for the given volume fraction of inclusions, the Young modulus of composite without interphase may not depend on the size of inclusions. It is because the classical continuum mechanics is scale-free. If the number of finite elements is sufficient, the fluctuations are caused by the insufficient number of particles.

The observed fluctuations are not so small. What is important is the difference between the obtained values and the Young modulus of the bulk PLA, 3000 MPa. This difference fluctuates from 128 MPa to 205 MPa.

ANSWER:

We have performed a lot of calculations, by FEM, increasing the number of elements, until a convergence could be achieved. The obtained deviations are rather low and do not alter the conclusions of this research.

SECOND Query

At the same time the values of the first column of Table 5 are inconsistent with the Figure 9c. In the figure, the interphase volume fractions are 0.025-0.05 for micrometer-size inclusions. However, in Table 5 the interphase volume fractions is 0.3 for the same micrometer-size inclusions. The interphase volume fraction 0.5 for analytical model for 0.5 μm particles in Table 5 is inconsistent with the Figure 10. One can see that 5-20 nm interphase thickness gives is good coincide with right point in the figure. The corresponding interphase volume fraction is below 0.01.

ANSWER:

The reviewer is right. There are some errors regarding the results in Table 5, when compared with Figure 9c. The calculations are correct, so the Figure 9c is correct. The interphase volume fraction values by FEA have been reported incorrectly in the Table. This has been now corrected. Also, the interphase volume fractions by the analytical model in Table 5 have been corrected. The calculated interphase thickness (as shown in Figure 10) was correct but the calculations for the

interphase volume fraction (due to miscalculation) were wrong. The text has been revised in accordance with the above corrections in the related calculations.

Regarding the last reviewer’s point, it is correct. As it is mentioned in the text the results by the analytical model are not further changed with changing the interphase thickness. The interphase volume fraction was found about 0.01.

All relative changes are highlighted in the revised text.

We hope our work is now worthy of publication in your esteemed Journal.

Best regards Dionysios Mouzakis Professor

Reviewer 2 Report

The authors have modified it as required and agreed to accept it.

Author Response

Thank you so much for your kind reply

Yours

D. Mouzakis

Round 3

Reviewer 1 Report

The manuscript have been revised, and all the questions have been answered. The manuscript can be published in the present form.